# Associations between objectively-measured and self-reported neighbourhood walkability on adherence and steps during an internet-delivered pedometer intervention

Anna Consoli[1], Alberto Nettel-Aguirre[1], John C. Spence[2], Tara-Leigh McHugh[2‡], Kerry Mummery[2‡], Gavin R. McCormack[1,3,4,5] *

1 Cumming School of Medicine, University of Calgary, Alberta, Canada, 2 Faculty of Kinesiology, Sport, and Recreation, University of Alberta, Alberta, Canada, 3 Faculty of Kinesiology, University of Calgary, Alberta, Canada, 4 School of Architecture, Planning and Landscape, University of Calgary, Alberta, Canada, 5 Faculty of Sport Sciences, Waseda University, Tokyo, Japan

⊚ These authors contributed equally to this work.
‡ These authors also contributed equally to this work.
* Gavin.McCormack@ucalgary.ca

**Data Availability Statement:** Data cannot be shared publicly because of making these data

## Abstract

### Background

Accumulating evidence suggests that the built environment is associated with physical activity. The extent to which the built environment may support adherence to physical activity interventions is unclear. The aim of this study was to investigate whether the neighbourhood built environment constrains or facilitates adherence and steps taken during a 12-week internet-delivered pedometer-based physical activity intervention (UWALK).

### Method

The study was undertaken in Calgary (Canada) between May 2016 and August 2017. Inactive adults (n = 573) completed a telephone survey measuring sociodemographic characteristics and perceived neighbourhood walkability. Following the survey, participants were mailed a pedometer and instructions for joining UWALK. Participants were asked to report their daily pedometer steps into the online program on a weekly basis for 12 weeks (84 days). Walk Score® estimated objective neighbourhood walkability and the Neighbourhood Environment Walkability Scale–Abbreviated (NEWS-A) measured participants self-reported neighbourhood walkability. Regression models estimated covariate-adjusted associations of objective and self-reported walkability with: 1) adherence to the UWALK intervention (count of days with steps reported and count of days with 10000 steps reported), and; 2) average daily pedometer steps.

### Results

On average, participants undertook 8565 (SD = 3030) steps per day, reported steps on 67 (SD = 22.3) of the 84 days, and achieved ≥10000 steps on 22 (SD = 20.5) of the 84 days.

publicly available is not covered by the current ethics approval (University of Calgary Conjoint Health Research Ethics Board; CHREB) and was not included as a condition in the original consent form agreed to by participants. Researchers not listed in the original ethics application (REB15-2944) cannot access confidential data. For more information please contact a CHREB representative (chreb@ucalgary.ca).

**Funding:** This work was supported by the Canadian Institutes of Health Research [MOP-142261, FDN-154331].The funders had no role in study design, data collection and analysis, decision to publish, or preparation of the manuscript.

**Competing interests:** The authors have declared that no competing interests exist.

Adjusting for covariates, a one-unit increase in self-reported walkability was associated on average with 45.76 (95CI 14.91, 76.61) more daily pedometer steps. Walk Score® was not significantly associated with steps. Neither objective nor self-reported walkability were significantly associated with the UWALK adherence outcomes.

## Conclusion

The neighbourhood built environment may support pedometer-measured physical activity but may not influence adherence to pedometer interventions. Perceived walkability may be more important than objectively-measured walkability in supporting physical activity during pedometer interventions.

## Introduction

Regular walking can assist adults in achieving levels of physical activity sufficient to obtain optimal health (i.e., 150 minutes/week of moderate-intensity physical activity) [1]. Walking is a no cost physical activity that has a low risk of injury [2, 3], can be undertaken by most able-bodied adults, can be incorporated into daily living (e.g., active transportation) [4], and is the preferred activity for inactive individuals initiating physical activity routines [5]. Regular walking provides health benefits such as increased physical fitness [6], reduced risk for cardiovascular disease [7], weight loss [8], improved blood pressure [9], and improved depressive symptoms [10]. Despite these potential health benefits, too few adults in North America [11, 12] and elsewhere [13] accumulate sufficient physical activity (including walking) for optimal health.

Several studies have investigated the impact of physical activity interventions, including pedometer-facilitated interventions, on walking [14–18]. Adults enrolled in pedometer interventions experience an average increase of physical activity of 26.9% from baseline which translates to an average of 2000 more steps per day [14, 19]. Furthermore, participation in pedometer interventions is associated with an average increase of 30–60 minutes of walking per week [20]. Pedometer interventions are effective at increasing physical activity among inactive adults [21], with people with the lowest baseline steps per day reporting the greatest increases in physical activity [22].

Given the growing popularity of pedometers for promoting physical activity, several studies have investigated the factors contributing to the effectiveness of pedometer-facilitated interventions [14, 19, 20]. Most of the success of pedometer interventions is attributed to strategies that increase user awareness and motivation, and thus behaviour modification (e.g., self-monitoring strategies and goal settings). Although rarely considered, the built environment may influence the success of physical activity interventions [23–25], including pedometer-facilitated interventions [26, 27].

Self-reported ("perceptions") [28–31] and objective [32–35] measures of the neighbourhood built environment are associated with physical activity. Neighbourhood features including street and sidewalk connectivity, residential density, proximity, mix of destinations and land uses, and pedestrian infrastructure are consistently associated with walking [36–42]. Higher objectively-measured walkability (e.g., higher Walk Score®) is positively associated with physical activity [43–45] and walking [46, 47]. Perceived neighbourhood features, including the presence of recreation facilities, sidewalks, shops and services and safety are also associated with physical activity [28–31]. Studies including both self-reported and objective

measures of the neighbourhood built environment often find stronger associations between perceptions and walking [48–50]. Qualitative study findings suggest that the built environment can be a barrier or facilitator in pedometer interventions [51, 52]; however, a dearth of quantitative evidence exists to support previous findings [26].

The aim of this study was to investigate whether the neighbourhood built environment constrains or facilitates physical activity during a 12-week internet-delivered pedometer-based intervention (UWALK) among adults. Specifically, we estimated the associations between objectively-measured walkability (Walk Score®) and self-reported walkability (Neighbourhood Environment Walkability Scale–Abbreviated [NEWS-A]), and: i) UWALK adherence; and ii) pedometer-measured physical activity.

## Methods

### Participants

This study involved a 12-week pedometer-based intervention (UWALK) as part of a one-group longitudinal quasi-experiment. Between May 2016 and August 2017, adult volunteers were recruited from 198 Calgary (Canada) neighbourhoods that belonged to a network of 147 community associations. Calgary is one of the major cities in Alberta, Canada. The average daily temperatures range from 16.5˚C in July to −6.8˚C in December. Winters are cold and the air temperature can drop below −30˚C [53].

Eligible participants included those who were at least 18 years of age, in the "contemplation" or "preparation" stages of physical activity behaviour change [54], not previously or currently enrolled in UWALK, reported no mobility issues preventing the proper use of a pedometer, and had internet access. To identify the stage of behaviour change, participants reported "true" or "false" to the following statements: 1) I currently do not participate in recreational or transportation-related physical activity; 2) I intend to participate in recreational or transportation-related physical activity in the next 3 months; 3) I am currently participating in recreational or transportation-related physical activity ≥3 days/week, and; 4) I have been participating in recreational or transportation-related physical activity ≥3 days/week for the past 6 months. Using a staging algorithm, contemplators responded true to statements 1 and 2 and preparers responded false to items 1 and 3 [55]. Only one adult per household was eligible to participate. Non-eligible individuals were directed to the UWALK website where they could join UWALK without being monitored as part of this study.

### Procedures

Community associations were approached to advertise the call for study participants via their newsletters, websites, and social media including Facebook and Twitter. Advertisements with community associations were posted for three months. Recruitment details were tweeted to members of the University of Calgary, City of Calgary, and Federation of Calgary Communities. Calls for study participation were also advertised in a free, widely distributed, local newspaper (Metro News). The call for participants listed the eligibility requirements for study participation and requested that interested adults email the research coordinator. Six-hundred individuals contacted the research coordinator. The research coordinator telephoned participants to confirm their study eligibility, described the study, obtained informed verbal consent, and where possible, administered a survey or scheduled the survey for a different time. The survey measured sociodemographic, perceptions of the neighbourhood walkability, and health information. The University of Calgary Conjoint Research Ethics Board approved this study (REB15-2944).

## Measures

**UWALK intervention adherence.** The definition of physical activity adherence varies widely across studies [56]. Studies have defined adherence as the percentage or total number of sessions attended, total duration (minutes) of physical activity participation, or percentage of data collected from self-reported questionnaires [56]. Despite these definitions, the measurement or operational definitions of physical activity intervention adherence are inconsistent, and no gold-standard exists [57]. Thus, we used UWALK website engagement as a source of data for estimating intervention adherence. Level of adherence was estimated from the count of days the participants entered their daily steps in the UWALK website (at least 84 days = the total days of UWALK intervention), and the count of days with 10000 steps or more. Achieving 10000 steps per day may be protective against depression [58], overweight and obesity [59, 60], and cardiometabolic risk factors [61]. Adults who accumulate more than 10000 steps per day are more likely to meet physical activity recommendations [1].

**Daily steps.** Participants were provided with a Piezo StepX pedometer which has demonstrated to be a reliable and valid measure of daily steps [62]. Written materials instructed participants to wear the pedometer on their hip and to wear the pedometer at all times except while sleeping, swimming, bathing, or engaging in contact sports. The instructions also requested participants to record their daily steps into the UWALK website for the entire 12 weeks. We provided participants with weekly step tracking sheets in case they were not able to enter their steps into the UWALK website daily. Participants could also record the flights of stairs climbed daily however, we excluded steps estimated based on stairs climbed (1 flight is equivalent to 10 steps), including only steps recorded by the pedometer. Based on previous studies [63], daily steps less than 100 and above 50000 were considered invalid and deleted. For each participant, we estimated mean daily steps for valid days only during the 12-week intervention.

**Neighbourhood walkability.** *Objectively-measured walkability*. A Walk Score® was linked to each participant's household via their 6-digit postal code. Walk Score® is a publicly available walkability index and reflects the level of access to nearby walkable amenities. Specifically, Walk Score® estimates neighbourhood walkability based on proximity to 13 amenity categories (i.e., grocery stores, coffee shops, restaurants, bars, movie theatres, schools, parks, libraries, book stores, fitness centres, drug stores, hardware stores, clothing/music stores) [64]. Walk Score® values range from 0 to 100 with low scores representing lower walkability and higher scores representing higher walkability. Walk Score® values less than 50 are considered car-dependent, while scores great than 90 are considered to be a Walker's paradise [65]. Walk Score® is correlated with other more comprehensive measures of walkability that capture are larger range of built features [66, 67]. Higher Walk Scores® are positively associated with walking and other physical activity [43–45, 66].

*Self-reported walkability*. The NEWS-A [68] measured participant's perceptions of the supportiveness of their neighbourhood for physical activity (neighbourhood defined as a 15-minute walk from home). The NEWS-A includes items that represented perceptions regarding neighbourhood residential density, connectivity, access to facilities and services, aesthetics, and safety. To ensure that the length of the telephone survey was manageable, only 24 out of 54 items, representing all domains, from the original NEWS-A were included in our survey. All items captured responses on a 4-point scale (i.e., "strongly disagree" to "strongly agree"). We used an established algorithm for creating a composite walkability index [69, 70], whereby lower scores represent less perceived walkability, and higher scores represent higher perceived walkability. The NEWS-A has acceptable reliability and validity [69], including a shorter version tested among Canadian adults [71, 72]. Our NEWS-A, with 24-items, had acceptable internal consistency in our sample (Cronbach's α = 0.80).

**Sociodemographic characteristics and weather.** During the survey, participants reported their age, sex, self-rated health (poor, fair, good, very good, or excellent), highest education achieved (high school diploma or less, college, vocation, or trade, university undergraduate, university postgraduate), annual gross household income (≤$39999, $40000 - $79999, ≥$80000, unknown/refused to answer), number of dependents ≤18 years of age at home, dog ownership (owner, non-owner), and motor vehicle availability for personal use (always/sometimes, never/do not drive). In addition, publicly available daytime temperature and daily precipitation data were collected and matched with the daily steps (Environment Canada— Calgary international airport) [53].

**UWALK intervention.** UWALK is an online multi-strategy, multi-sector, theory-informed, community-wide approach intervention (www.uwalk.ca) to promote physical activity in Alberta, Canada [52]. UWALK was modelled on other pedometer-based interventions that have successfully increased physical activity [73, 74]. The primary focus is on accumulation of daily steps and flights of stairs (10 steps/stairs are equivalent to 1 flight). Participants are encouraged to use electronic devices to self-monitor their physical activity (e.g., pedometers, smartphone applications). UWALK includes a website where participants record their pedometer steps and track their own progress. In addition, the UWALK intervention uses simple but established health promotion approaches for empowering individuals to walk as a mean of increasing their physical activity levels [52]. For this study we used the existing UWALK promotional material and online infrastructure. Upon completion of the survey, a study package was sent to the participant's residence. The package contained the pedometer, instructions on how to use and wear the pedometer, and instructions for the UWALK website (i.e., how to register and track physical activity), a daily tracking sheet, and the UWALK promotional material.

## Statistical analysis

We summarized data using means, standard deviations or frequencies. We used Pearson's chi-squares (for categorical variables) and independent t-tests (for continuous variables) to identify differences in sociodemographic and built environment characteristics of those who did with those who did not register in the UWALK intervention after the survey was completed. For all participants, we compared the first and last reported week of average daily steps using a dependent sample t-test. Using a dependent sample t-test, we also compared the first week and the last week of average daily steps for UWALK participants who entered steps each week of the 12-week intervention.

We estimated the associations of objective neighbourhood walkability (Walk Score®) and self-reported neighbourhood walkability (NEWS-A) with UWALK days of adherence (negative binomial regression), days achieving ≥10000 steps (negative binomial regression), and daily steps (linear regression). For the count of days with ≥10000 steps, individual's total days were specified as an offset variable to model the count of days with ≥10000 steps (count over the total days of steps of each participant). Two separate models were fitted to estimate the effect of objective neighbourhood walkability and self-reported neighbourhood walkability on each outcome of adherence, and physical activity, followed by a final model that included both objective and self-reported neighbourhood walkability. We planned to use the negative binomial regression if Poisson count data were over dispersed (variance larger than the mean). From these models we obtained measures of association between walkability and outcomes: Odds Ratios (ORs; logistic regression); unstandardized beta coefficients (bs; linear regression); and Incidence Rate Ratios (IRRs; negative binomial regression). We checked assumptions for all models (e.g., linearity, independence, normality, and homoscedasticity). To assess

collinearity between self-reported and objective measures of walkability, we studied the Pearson correlation coefficient before model fitting and the variance inflation factor of the model including both independent variables. We adjusted regression models for all sociodemographic and weather variables. Statistical significance level was set at alpha of 0.05 and we reported 95 percent confidence intervals (95CI) for each measure of association. Stata version 13.0 (Stata Corp, TX) was used to conduct the analyses.

## Results

### Sample characteristics

Complete data were available for $n$ = 573 participants, of whom $n$ = 466 registered in UWALK ($n$ = 107 eligible participants did not register after completing the survey). Except for annual gross household income ($p$ = 0.02), those who did and did not register in UWALK were not significantly different on all other characteristics (Table 1). Those who registered in UWALK were on average 49.15 years old (SD = 14.40). Of these, 83% were women, 45% were in good health, 40% received university education, 32% had annual gross household income ≥$80000, had on average 0.71 child ≤18 years old at home (SD = 1.07), 79% were not dog owners, and 91% had access to a motor vehicle.

The mean (SD) Walk Score® and NEWS-A score among those registered was 44.66 (21.30) and 77.13 (8.90), respectively (Table 1). The lowest Walk Score® was 2 and the highest

**Table 1. Sociodemographic and built environment characteristics for participants who registered in UWALK and participants who did not register in UWALK.**

| Characteristics | Category | Study participants (n = 466) | Did not register (n = 107) | p value |
|---|---|---|---|---|
| | | Mean (SD) | Mean (SD) | |
| Age in years | = = | 49.15 (14.40) | 50.11 (14.57) | 0.53 |
| Sex % | Female | 83.05 | 77.57 | 0.18 |
| Self-rated health % | Poor | 3.86 | 8.41 | 0.07 |
| | Fair | 23.61 | 31.78 | |
| | Good | 44.85 | 37.38 | |
| | Very good | 23.82 | 17.76 | |
| | Excellent | 3.86 | 4.67 | |
| Highest education completed % | High school diploma or less | 15.02 | 17.76 | 0.92 |
| | College, vocation, or trade | 24.25 | 23.36 | |
| | University undergraduate | 40.13 | 38.32 | |
| | University postgraduate | 20.60 | 20.56 | |
| Annual gross household income % | ≤$39999 | 13.09 | 16.82 | 0.02* |
| | $40000 - $79999 | 18.45 | 29.91 | |
| | ≥$80000 | 32.19 | 24.30 | |
| | Unknown | 36.27 | 28.97 | |
| Number of dependents ≤18 years old | = = | 0.71 (1.07) | 0.78 (1.16) | 0.58 |
| Dog owner % | Yes | 21.03 | 16.82 | 0.33 |
| | No | 78.97 | 83.18 | |
| Motor vehicle available for personal use % | Always/Sometimes | 91.20 | 94.39 | 0.28 |
| | Never/Do not drive | 8.80 | 5.61 | |
| Walk Score® | = = | 44.66 (21.30) | 44.28 (19.48) | 0.87 |
| NEWS-A[a] | = = | 77.13 (8.98) | 75.98 (9.67) | 0.24 |

Note: Independent t-test was used for continuous variables. Pearson Chi-square test was used for categorical variables.

[a] The abbreviated Neighbourhood Environment Walkability Scale (NEWS-A).

* < .05; b: unstandardized.

was 98 (possible range 0–100). The lowest NEWS-A score was 38 and the highest was 96 (possible range 24–96). Walk Score® and NEWS-A score were correlated (r = 0.17, *p* = 0.001) and low level of collinearity was present (VIF = 1.00). The mean (SD) 24-hour precipitation and temperature was 1.06 mm (0.72) and 3.62˚C (8.50), respectively. The majority of the participants initiated UWALK between September 2016 (late summer) and May 2017 (mid spring).

## Neighbourhood walkability and UWALK adherence

On average, participants entered steps in UWALK on 67.2 (SD = 22.3) days of the 84 days of the intervention. Adjusting for all covariates, Walk Score® and the NEWS-A score were not significantly associated with count of days steps were entered in UWALK (Table 2).

**Table 2. Associations between objectively-measured walkability (Walk Score®) and self-reported walkability (NEWS-A) and UWALK adherence and pedometer-measured physical activity.**

|  | UWALK adherence days with steps | UWALK adherence days with 10,000 steps | UWALK pedometer-measured physical activity |
|---|---|---|---|
|  | (n = 466) | (n = 454)[b] | (n = 466) |
|  | IRR (95CI) | IRR (95CI) | b (95CI) |
| Walk Score® | 1.00 (0.99, 1.00) | 1.00 (0.99, 1.00) | 3.98 (-8.98, 16.94) |
| NEWS-A | 1.00 (0.99, 1.00) | 1.01 (1.00, 1.02) | 45.76 (14.91, 76.61)* |
| Age in years | 1.00 (0.99, 1.00) | 1.00 (1.00, 1.01) | 2.78 (-17.91, 23.47) |
| Sex (*ref: Female*) | 1.02 (0.92, 1.14) | 0.92 (0.71, 1.19) | 41.22 (-677.47, 759.93) |
| Self-rated health (*ref: Poor*) |  |  |  |
| Fair | 1.09 (0.87, 1.36) | 1.37 (0.80, 2.32) | 847.40 (-620.63, 2315.43) |
| Good | 1.22 (0.99, 1.52) | 1.90 (1.13, 3.19)* | 1354.58 (-68.77, 2777.92) |
| Very good | 1.17 (0.94, 1.47) | 1.55 (0.90, 2.66) | 1110.53 (-374.93, 2595.99) |
| Excellent | 1.11 (0.83, 1.49) | 2.09 (1.03, 4.23)* | 2262.97 (332.93, 4193.01)* |
| Highest education completed (*ref: High school or less*) |  |  |  |
| College, vocation, or trade | 1.08 (0.94, 1.23) | 0.98 (0.70, 1.38) | -492.08 (-1380.99, 396.83) |
| University undergraduate | 1.05 (0.93, 1.19) | 0.96 (0.71, 1.31) | -527.00 (-1355.29301.29) |
| University postgraduate | 1.08 (0.94, 1.24) | 0.88 (0.62, 1.26) | -439.63 (-1372.16, 492.90) |
| Annual gross household income (*ref: ≤$39999)* |  |  |  |
| $40000 - $79999 | 0.93 (0.80, 1.08) | 0.71 (0.49, 1.03) | -623.58 (-1606.74, 359.57) |
| ≥$80000 | 0.99 (0.86, 1.13) | 1.05 (0.73, 1.49) | 23.57 (-902.592, 949.73) |
| Unknown | 0.96 (0.84, 1.09) | 1.04 (0.74, 1.44) | 239.12 (-646.47, 1124.72) |
| Number of dependents ≤18 years old at home | 1.02 (0.98, 1.06) | 1.11 (1.00, 1.23)* | 379.44 (108.71, 650.18)* |
| Dog owner (*ref: non-owner*) | 0.93 (0.84, 1.03) | 1.01 (0.79, 1.29) | 698.95 (30.09, 1367.82)* |
| Motor vehicle available (*ref: Never/do not drive*) | 0.88 (0.75, 1.02) | 0.59 (0.41, 0.87)* | -1368.86 (-2393.32, 344.41)* |
| Daily mean temperature (Celsius)[c] | 1.00 (0.99, 1.00) | 1.02 (1.01, 1.04)* | 48.43 (25.66, 71.21)* |
| Daily mean total precipitation (mm)[d] | 1.00 (0.99, 1.00) | 0.90 (0.71, 1.13) | 4.68 (-83.09, 92.44) |
| Intercept |  |  | 4654.33 (1638.52, 7670.14) |

[a] Four missing data excluded from the analysis

[b] Twelve missing data excluded from the analysis.

[c] Mean temperature was based on the 12 weeks UWALK intervention for each participant.

[d] Mean total precipitation refers to rain and snow.

Odd Ratio (OR), Incidence Rate Ratio (IRR), Beta coefficient (b): Unstandardized; 95CI: 95 percent confidence interval

*$p$ < .05; All models adjusted for all sociodemographic characteristics and weather.

Furthermore, none of the covariates were significantly associated with count of days steps were entered in UWALK. On average, participants reported achieving ≥10000 steps on 22.5 (SD = 20.5) days of the 84 days UWALK intervention. Adjusting for all covariates, neither Walk Score® nor the NEWS-A score was significantly associated with count of days achieving ≥10000 steps (Table 2). In the fully-adjusted model, good and excellent self-rated health (compared to poor health; IRR = 1.9; 95CI 1.1, 3.2, *p* = 0.02, IRR = 2.1; 95CI 1.0, 4.2, *p* = 0.04), number of dependents ≤18 years old (IRR = 1.1; 95CI 1.0, 1.2, *p* = 0.04), access to a motor vehicle (IRR = 0.6; 95CI 0.4, 0.9, *p* = 0.01), and daily mean temperature (IRR = 1.0; 95CI 1.1, 1.0, *p* = 0.01) were associated with count of days achieving ≥10000 steps (Table 2).

## Neighbourhood walkability and pedometer-determined physical activity

On average, participants reported undertaking 8565 (SD = 3030) steps per day during the UWALK intervention. The differences between the average daily steps undertaken in the first and last week of the UWALK intervention were not statistically significant for those who entered step data all weeks (8634.47 vs. 8896.69, t = -1.13, *p* = 0.26, n = 216), and those who did not enter step data all weeks (8290.91 vs. 8268.46, t = 0.11, *p* = 0.92, n = 250) during the 12 week UWALK intervention. Adjusting for all covariates, NEWS-A score (b = 45.8; 95CI 14.9, 76.6, *p* = 0.004) but not Walk Score® (b = 3.9; 95CI -8.9, 16.9, *p* = 0.5) was associated with mean daily pedometer steps (Table 2). In the fully-adjusted model, excellent self-rated health (compared to poor health; b = 2262.9; 95CI 332.9, 4193.0, *p* = 0.02), number of dependents ≤18 years old (b = 379.4; 95CI 108.7, 650.1, *p* = 0.01), dog ownership (b = 698.9; 95CI 30.0, 1367.8, *p* = 0.04), access to a motor vehicle (b = -1368.8; 95CI -2393.3, -344.4, *p* = 0.01) and daily mean temperature (b = 48.4; 95CI 25.6, 71.2, *p* = 0.001) were associated with mean daily pedometer steps (Table 2).

## Discussion

We examined the effects of the self-reported and objectively-measured neighbourhood built environment on physical activity during a 12-week internet-delivered pedometer-based intervention. Our findings show that a one-unit increase in self-reported walkability was associated on average with 46 more daily steps. Conversely, the objectively-measured neighbourhood walkability was not associated with steps during the intervention. Self-reported and objectively-measured neighbourhood walkability were also not associated with adherence to the UWALK intervention. Furthermore, the steps measured in the first and last week of the intervention for each participant were not significantly different.

Our finding of a positive association for perceived walkability and no significant association for objectively-measured walkability is consistent with other studies [75, 76]. Perception of the built environment appears to be more strongly related to behaviour change than objectively-measured built environment characteristics [48, 77, 78]. In a study undertaken in Japan [75], adults who reported a positive perception of the neighbourhood were almost twice as likely to engage in leisure walking compared to those who reported a negative perception of the neighbourhood. However, objective walkability was not associated with leisure walking. Similarly, among US adults, perceived walkability was associated with 12 more minutes of walking per week while Walk Score® was not related to walking [76]. Notably, similarly defined perceived and objective neighbourhood characteristics have low-to-moderate agreement [49, 77], which suggests that these measures should not be used interchangeably [79]. In our study, the NEWS-A and Walk Score® were weakly correlated suggesting they are likely measuring different aspects of neighbourhood walkability and may influence walking in different ways [80, 81]. Future research should explore the effects of objectively-measured and self-reported

individual neighbourhood built features (e.g., connectivity, density, land use and destination proximity and mix, pedestrian infrastructure, and safety) in relation to the effectiveness of pedometer interventions.

The stronger association of the self-reported walkability and daily pedometer steps compared with objectively-measured neighbourhood walkability and daily pedometer steps might reflect that the type of walking UWALK participants undertook. Other studies have found that some perceived features (e.g., safety and aesthetics) are related to leisure walking [36, 82] while objective walkability tends to be associated with transportation walking [83]. Our participants might have accumulated much of their steps through leisure walking. This is somewhat supported by qualitative findings from follow-up with UWALK participants, although transportation walking was also mentioned for accumulating steps [84]. Furthermore, we used Walk Score® to estimate the neighbourhood walkability. Although Walk Score® is a valid measure of accessibility to nearby amenities in urban neighbourhoods, a major limitation is that it does not account for built environment characteristics such as aesthetics, safety or presence of physical activity facilities, which are often perceived as important influences of leisure-time walking [85]. Conversely, these findings could challenge the assumption of most ecological models that the environment has direct influences on behaviour [86, 87]. Instead, it may be that the effects of the environment are mediated by perceptions of the individual which would be consistent with social cognitive explanations [88].

Living in a high walkable neighbourhood and having a positive perception of the neighbourhood did not appear to contribute to more days of walking or to a high number of days with 10000 steps among adults participating in the UWALK intervention. Our findings are inconsistent with other studies that reported positive associations between environmental factors and adherence to a physical activity intervention. Findings from a cross-sectional study [24] found that neighbourhood aesthetic and satisfaction with the ease and pleasantness of the neighbourhood was positively associated with more vigorous physical activity and with 30% more participants achieving the physical activity recommendations. Similarly, in a quasi-experimental study, the objectively-measured presence of public recreation centres and/or shopping malls (one or both) was associated with greatest adherence (percentage of prescribed walks completed) to a walking intervention among African American women [25]. However, these studies only examined the self-reported or the objectively-measured built environment separately, in relation to physical activity. On the contrary, Sugiyama et al. [89] found that the perceived and the objective presence of more green space in the neighbourhood was associated with a higher likelihood of maintaining recreational walking over four years. In our study, other built characteristics might have influenced the adherence to UWALK. Specifically, inclement weather or unfavorable outdoor conditions (e.g., ice on the ground) might have been perceived as a barrier to daily walking which resulted in less frequent walking or walks of shorter duration. The negative impact of weather on physical activity has been observed in other studies using pedometer-based interventions [90, 91], which reported lower counts of steps in winter compared to other seasons. However, strategies can be adopted to increase adherence to a physical activity intervention. For example, Heesch et al. [92] describes how participants who were not achieving the recommended levels of physical activity, requested information from the program staff on how to cope with poor weather and how to obtain information on places to walk in their community. The impact of weather on steps might also depend on geographical location. Congruent with other Canadian studies [93], we found a positive linear association between temperature and steps however, in other locations (e.g., Japan), others have found non-linear relationships between temperature and steps [94, 95].

This study has several limitations. Participants self-selected to participate, and the majority were middle-aged, highly educated women with medium to high household incomes.

Sociodemographic characteristics of volunteers might be different from those who do not volunteer for research studies [96], thus limiting the generalizability of our findings. Participants might have walked in locations outside their neighbourhood or accumulated their steps through activities inside their homes, which could attenuate associations between neighbourhood walkability, and steps. Our quasi-experiment did not include a control group and we found no difference in average daily steps undertaken earlier versus later in UWALK, thus it remains unclear whether UWALK, independent of the built environment, affected physical activity. It is also unclear the extent to which UWALK and the built environment might be associated with adherence and steps over a longer intervention period. The accuracy of participant reporting of steps in the UWALK website is unknown. We used a 24-item version of the NEWS-A that had high internal consistency but which may differ from the original NEW-A in terms of its content and predictive validity.

A strength of this study was the quasi-experimental design that included capturing self-reported and objective neighbourhood walkability data prior to participants beginning UWALK. However, it is possible that the perceptions of neighbourhood walkability among participants might have changed as a result of their involvement in the UWALK intervention [97]. Other strengths include the inclusion of multiple measures of adherence and behaviour, inclusion of objectively measured physical activity using pedometers, inclusion of self-reported and objectively measured walkability, and recruitment of inactive adults.

## Conclusion

Our study provides evidence suggesting that the neighbourhood built environment may affect individual-targeted interventions, like UWALK, and influence on physical activity. Perceptions of neighbourhood walkability, but not objectively measured walkability, appear to be important for supporting the number of steps taken among inactive adults participating in an internet-facilitated pedometer intervention. To increase daily steps, strategies targeting the individual's perceptions of the neighbourhood (e.g., provision of maps with walkable routes, suggestions about community recreations events) should be considered when designing physical activity interventions within different neighbourhood contexts. Given that neighbourhood walkability was not associated with UWALK adherence might suggest that other non-environment strategies are needed to encourage uptake of physical activity in community-based interventions.

## Acknowledgments

We acknowledge Rosemary Perry, Rhianne Fiolka and Anita Blackstaffe for their support in recruitment, data collection, and data management. We also thank Cally Jennings for the UWALK data coordination.

## Author Contributions

**Conceptualization:** Alberto Nettel-Aguirre, John C. Spence, Tara-Leigh McHugh, Kerry Mummery, Gavin R. McCormack.

**Data curation:** Gavin R. McCormack.

**Formal analysis:** Anna Consoli.

**Funding acquisition:** Alberto Nettel-Aguirre, John C. Spence, Tara-Leigh McHugh, Kerry Mummery, Gavin R. McCormack.

**Investigation:** Anna Consoli, John C. Spence, Tara-Leigh McHugh, Kerry Mummery, Gavin R. McCormack.

**Methodology:** Anna Consoli, Alberto Nettel-Aguirre, John C. Spence, Tara-Leigh McHugh, Kerry Mummery, Gavin R. McCormack.

**Supervision:** Gavin R. McCormack.

**Writing – original draft:** Anna Consoli, Alberto Nettel-Aguirre, John C. Spence, Tara-Leigh McHugh, Kerry Mummery, Gavin R. McCormack.

**Writing – review & editing:** Anna Consoli, Alberto Nettel-Aguirre, John C. Spence, Tara-Leigh McHugh, Kerry Mummery, Gavin R. McCormack.

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
