## [Decision Letter · Decision Letter 0]

18 Sep 2020

PONE-D-20-20633

Associations between objectively-measured and self-reported neighbourhood walkability on adoption, adherence, and steps during an internet-delivered pedometer intervention

PLOS ONE

Dear Dr. McCormack,

Thank you for submitting your manuscript to PLOS ONE. After careful consideration, we feel that it has merit but does not fully meet PLOS ONE’s publication criteria as it currently stands. Therefore, we invite you to submit a revised version of the manuscript that addresses the points raised during the review process.

We look forward to receiving your revised manuscript.

Kind regards,

Anne Vuillemin

Academic Editor

PLOS ONE

Journal Requirements:

Reviewers' comments:

Reviewer's Responses to Questions

**Comments to the Author**

1. Is the manuscript technically sound, and do the data support the conclusions?

Reviewer #1: Partly

Reviewer #2: Yes

2. Has the statistical analysis been performed appropriately and rigorously? 

Reviewer #1: Yes

Reviewer #2: Yes

3. Have the authors made all data underlying the findings in their manuscript fully available?

Reviewer #1: No

Reviewer #2: No

4. Is the manuscript presented in an intelligible fashion and written in standard English?

Reviewer #1: Yes

Reviewer #2: Yes

5. Review Comments to the Author

Reviewer #1: This study investigated whether the neighbourhood built environment constrains or facilitates adoption, adherence, and steps taken during a 12-week internet-delivered pedometer-based physical activity intervention. The topics covered are interesting and worthy of constructive discussion. However, some revisions are required before accepting the manuscript.

Major comments

1. Please add the significance of checking "intervention adoption". Why did you threshold 6 days from the telephone survey? Given the difference in the number of days before participants received the pedometer and the difference in their skill to use the UWALK website, I doubt that this was a suitable target variable.

2. You discussed as if you were comparing perceived and objective walkability by comparing NEWS-A and Walk Score, but the range of built environment characteristics measured by the two index is different. If you want to compare with objective walkability, you should see individual items of NEWS-A instead of composite score.

Minor comments

L.171

Please specify how many people were contemplators, prepares, and non-eligible.

L.245

Previous studies showed that the relationship between temperature and step count was not linear (step count decreased when temperature was too high). Even if you don't need to consider it in your target area and season, you should excuse it with reference to such studies.

L. 251

Please explain how you used flights of stairs.

L. 334

Please add how you judged whether they "completed" or not.

Reviewer #2: This is an interesting, well-conducted and well-written study and I have few comments that need addressing.

The early vs late adoption dichotomisation seems meaningless to me. Six days after the survey one is ‘early, after 7 days one is ‘late’…though it’d be hard to tell the difference. I recommend a gap instead: for example, if one gets involved within one week, then they are early, if they still haven’t started after 3 weeks, then they are late…that would be more meaningful too me.

Also, I think the idea of early/late adoption refers to the diffusion of innovation theory…however, I’m not sure it applies in this circumstance: early adopters were for example those who were the first to buy a smartphone when they were first introduced to the market, and late adopters are those who only got one after 80% of the population already had one… The sort of adoption you are referring to is something completely different. I think assessing whether there was adoption (yes/no), is more meaningful over when it happened (early/late), so I would recommend to remove this from the paper. It is not surprising to see no difference in objective and subjective walkability based on this variable.

As only 24 out of 54 items of the original NEWS-A were included I would say that any reference to the reliability and validity of this measure is meaningless, this should be acknowledged as a limitation.

The first sentence of the conclusion is too strongly worded: this study didn’t find evidence of the build environment being a moderator; only perceptions of that neighbourhood had an influence.

6. PLOS authors have the option to publish the peer review history of their article (what does this mean?). If published, this will include your full peer review and any attached files.

Reviewer #1: No

Reviewer #2: No

---

## [Author Response · Author response to Decision Letter 0]

19 Oct 2020

Reply to Reviewers

Reviewer #1: 

1. This study investigated whether the neighbourhood built environment constrains or facilitates adoption, adherence, and steps taken during a 12-week internet-delivered pedometer-based physical activity intervention. The topics covered are interesting and worthy of constructive discussion. However, some revisions are required before accepting the manuscript.

We thank the reviewer for their positive comment regarding our manuscript and we appreciated the feedback provided for improving the manuscript.

Major comments

2. Please add the significance of checking "intervention adoption". Why did you threshold 6 days from the telephone survey? Given the difference in the number of days before participants received the pedometer and the difference in their skill to use the UWALK website, I doubt that this was a suitable target variable.

The reviewer 1 raises an important point, one that reviewer 2 also mentioned. Reviewer 2, suggested we remove the “adoption” outcome from the manuscript. Given the limitations raised by both reviewers regarding the “adoption” variable, we have decided to remove all reference to this variable from the revised manuscript. 

3. You discussed as if you were comparing perceived and objective walkability by comparing NEWS-A and Walk Score, but the range of built environment characteristics measured by the two index is different. If you want to compare with objective walkability, you should see individual items of NEWS-A instead of composite score.

We agree with the reviewer. The NEWS-A and Walk Score® are likely capturing some but also different built environment features that may support walking and other physical activity. Notably, Walk Score® is correlated with other more comprehensive measures of objectively-measured walkability (references 66 and 67 cited on page 9 of revised manuscript) suggesting that it may indirectly reflect a wider range of built features in addition to those features included in Walk Score® operational definition. We include in the revised manuscript the estimated correlation between the NEWS-A total walkability score and the Walk Score® (r=0.17) to demonstrate they are indeed weakly related (page 12). Examining the associations between the individual NEWS-A items and the pedometer related outcomes is beyond the scope of the manuscript but we thank the reviewer for making us aware of this potential new avenue of investigation. In response to this comment, we now mention the limitations regarding the associations between of the NEWS-A and Walk Score® including their differences in content validity in the limitations section (pg. 19, lines 355-360 of tracked version).

Minor comments

4. [L.171] Please specify how many people were contemplators, prepares, and non-eligible.

600 hundred participants contacted the research coordinator and underwent eligibility screening. We do not have a record with the breakdown of characteristics for those excluded from the study at the screening stage. Note the recruitment material included details related to the inclusion and exclusion criteria for the study, thus the volunteers are somewhat bias. Presenting information regarding the breakdown by contemplators, preparers, and others is unlikely to be meaningful or representative. In response to this reviewers comment, we have added information about the total number screened in the revised manuscript (line 178, page 7). 

5. [L.245] Previous studies showed that the relationship between temperature and step count was not linear (step count decreased when temperature was too high). Even if you don't need to consider it in your target area and season, you should excuse it with reference to such studies.

We now mention and cite two published studies demonstrating non-linear relationships between temperature and steps (lines 401-404, page 21 of tracked version).

6. [L. 251] Please explain how you used flights of stairs.

UWALK promotes the use of stairs and flights of stairs climbed were captured (line 249-250, page 10), however, we did not convert flights climbed to steps in our study (i.e., 10 steps/stairs are equivalent to 1 flight). In the revised manuscript, we now include a sentence describing the exclusion of stair-estimated steps from the step count (Line 206-208, page 8).

7. [L. 334] Please add how you judged whether they "completed" or not.

In response the reviewer comment, we have modified the following sentences to provide clarity: “Using a dependent sample t-test, we also compared the first week and the last week of average daily steps for UWALK participants who entered steps each week of the 12 week intervention” (lines 266-268, page 11; tracked version) and “The differences between the average daily steps undertaken in the first and last week of the UWALK intervention were not statistically significant for those who entered step data all weeks (8634.47 vs. 8896.69, t = -1.13, p = 0.26, n = 216), and those who did not enter step data all weeks (8290.91 vs. 8268.46, t = 0.11, p = 0.92, n = 250) during the 12 week UWALK intervention.” (lines 324-328, page 17; tracked version).

Reviewer #2: 

This is an interesting, well-conducted and well-written study and I have few comments that need addressing.

1. The early vs late adoption dichotomisation seems meaningless to me. Six days after the survey one is ‘early, after 7 days one is ‘late’…though it’d be hard to tell the difference. I recommend a gap instead: for example, if one gets involved within one week, then they are early, if they still haven’t started after 3 weeks, then they are late…that would be more meaningful too me. Also, I think the idea of early/late adoption refers to the diffusion of innovation theory…however, I’m not sure it applies in this circumstance: early adopters were for example those who were the first to buy a smartphone when they were first introduced to the market, and late adopters are those who only got one after 80% of the population already had one… The sort of adoption you are referring to is something completely different. I think assessing whether there was adoption (yes/no), is more meaningful over when it happened (early/late), so I would recommend to remove this from the paper. It is not surprising to see no difference in objective and subjective walkability based on this variable.

Given the limitations regarding the adoption variable raised by both reviewers, as suggested by reviewer 2, we have removed all reference to the “adoption” variable from the revised manuscript. 

2. As only 24 out of 54 items of the original NEWS-A were included I would say that any reference to the reliability and validity of this measure is meaningless, this should be acknowledged as a limitation.

We now mention the reliability of short versions of the NEWS-A in the methods and note that our 24-items NEWS-A has acceptable internal consistency (lines 233-236; tracked version). We have also added the following sentence to the limitations (lines 417-419, page 21; tracked version): “We used a 24-item version of the NEWS-A that had high internal consistency but which may differ from the original NEW-A in terms of its content and predictive validity.”

3. The first sentence of the conclusion is too strongly worded: this study didn’t find evidence of the build environment being a moderator; only perceptions of that neighbourhood had an influence.

We have revised this sentence (lines 428-429, page 22; tracked version): “Our study provides evidence suggesting that the neighbourhood built environment may affect individual-targeted interventions, like UWALK, and influence on physical activity.”

Reply to editorial comments

a. Please clarify the sources of funding (financial or material support) for your study. List the grants or organizations that supported your study, including funding received from your institution.

d. If you did not receive any funding for this study, please state: “The authors received no specific funding for this work.”

We have added information to the cover letter to the editor as requested.

---

## [Editor Report · Decision Letter 1]

13 Nov 2020

Associations between objectively-measured and self-reported neighbourhood walkability on adherence and steps during an internet-delivered pedometer intervention

PONE-D-20-20633R1

Dear Dr. McCormack,

We’re pleased to inform you that your manuscript has been judged scientifically suitable for publication and will be formally accepted for publication once it meets all outstanding technical requirements.

Kind regards,

Anne Vuillemin

Academic Editor

PLOS ONE
---

## [Editor Report · Acceptance letter]

24 Nov 2020

PONE-D-20-20633R1 

Associations between objectively-measured and self-reported neighbourhood walkability on adherence and steps during an internet-delivered pedometer intervention 

Dear Dr. McCormack:

I'm pleased to inform you that your manuscript has been deemed suitable for publication in PLOS ONE. Congratulations! Your manuscript is now with our production department. 

Kind regards, 

on behalf of

Dr. Anne Vuillemin 

Academic Editor

PLOS ONE